# Prolonged COVID-19 Infection in a Patient with Complete Remission from Follicular Lymphoma with Hyperosmolar Hyperglycemic Syndrome

**DOI:** 10.3390/geriatrics8060110

**Published:** 2023-11-12

**Authors:** Takuya Omura, Akira Katsumi, Shuji Kawashima, Masahiro Naya, Haruhiko Tokuda

**Affiliations:** 1Department of Endocrinology and Metabolism, National Center for Geriatrics and Gerontology, 7-430 Morioka-cho, Obu 474-8511, Aichi, Japan; 2Department of Metabolic Research, Research Institute, National Center for Geriatrics and Gerontology, 7-430 Morioka-cho, Obu 474-8511, Aichi, Japan; 3Department of Hematology, National Center for Geriatrics and Gerontology, 7-430 Morioka-cho, Obu 474-8511, Aichi, Japan; 4Department of Clinical Laboratory, National Center for Geriatrics and Gerontology, 7-430 Morioka-cho, Obu 474-8511, Aichi, Japan

**Keywords:** follicular lymphoma, hyperosmolar hyperglycemia syndrome, prolonged COVID-19, tixagevimab/cilgavimab

## Abstract

An 81-year-old woman undergoing B-cell depletion therapy developed COVID-19 and a hyperglycemic hyperosmotic state. She had a history of multiple vaccinations against coronaviruses but had persistent antigen positivity. Strategies to prevent the development of COVID-19 in immunosuppressed patients have not been established. Moreover, there is no standard treatment for prolonged antigen positivity. In this case, we were able to follow IgG antibodies during the course of treatment. The absence of N-IgG antibody titer elevation despite an effective immune response triggered by the vaccine is of great interest. The impaired humoral response observed in patients with lymphoma after anti-CD20 treatment implies the need for a justified different vaccination strategy for these patients. Furthermore, negative N-IgG titers in the immunosuppressed state may serve as an indicator of resistance to therapy.

## 1. Introduction

Patients undergoing B-cell depletion therapy, such as rituximab, have been shown to have a lower likelihood of developing elevated antibody titers [1]. In the following presented case, the S-IgG levels were elevated because of vaccination and infection, whereas the N-IgG levels remained low, despite the typical increase observed after COVID-19 infection. This impaired N-IgG titer is believed to have contributed to the prolonged duration of COVID-19 infection.

The absence of N-IgG antibody titer elevation despite an effective immune response triggered by the vaccine is of great interest. To the best of our knowledge, the immune response to this phenomenon has not yet been reported, and the optimal treatment approach for immunosuppressed hematopoietic patients with prolonged COVID-19 infection remains undetermined. Furthermore, there is limited understanding of the immune response to tixagevimab and cilgavimab (Evusheld) in hematopoietic patients undergoing B-cell depletion therapy. We present this case because it provides novel insights into these unresolved issues.

## 2. Case Presentation

An 81-year-old woman visited the emergency room complaining of nausea, vomiting, and abdominal pain. She had type 2 diabetes mellitus, gastric cancer (postoperative), and follicular lymphoma. Her lymphoma was treated with chemotherapy (rituximab plus bendamustine) when she was aged 78 years, and she achieved complete remission, followed by 12 cycles of rituximab maintenance therapy.

At the time of the visit, the patient’s vital signs were as follows: body temperature, 36.8 °C; blood pressure, 134/95 mmHg; heart rate, 112 bpm without arrhythmia, and SpO_2_, 97% (room air). Blood tests revealed that the glucose (Glu) and plasma osmolality levels were 437 mg/dL and 309.8 mOsm/L, respectively, and the pH and arterial blood HCO_3_^−^ level were 7.66 and 12.3 mEq/L, respectively. Other major blood laboratory findings were as follows: white blood cell (WBC), 2200/μL; hemoglobin (Hb), 11.9 g/dL; C-reactive protein (CRP), 1.78 mg/dL; glycated haemoglobin (HbA1c), 8.1%; and creatinine-based estimated glomerular filtration rate (eGFRcr), 46.3 mL/min/1.73 m^2^.

We diagnosed her with hyperosmolar hyperglycemic syndrome (HHS) and decided to hospitalize her. She was rehydrated with saline solution and administered insulin intravenously. She had received four doses of the corona vaccine and anti-SARS-CoV-2 monoclonal antibodies (tixagevimab and cilgavimab [Evusheld]) 3 months earlier, as immunocompromised patients have significantly lower seroconversion rates from vaccination. She underwent a SARS-CoV-2 antigen quantification test (Sysmex, Kobe, Japan), which revealed a positive result (30,802 cutoff index [COI]). She was thus diagnosed with a COVID-19 infection. Fortunately, the respiratory status remained stable, and a CT chest scan showed no evidence of pneumonia, indicating that she was still in moderate I of Japanese classification for severity. Regarding her risk of severity in terms of hyperglycemia and lymphoma, despite complete remission, Remdesivir was administered, because severe nausea and vomiting symptoms made it difficult for her to take oral medications. In addition, an antimicrobial (Cefepime 1 g/day) was administered to avoid bacterial infection of the respiratory tract.

After the improvement of hyperglycemia, insulin was switched to subcutaneous injection. On day 7 after admission, the corona antigen remained positive (1640 COI), but the value had remarkably decreased, indicating an obvious trend toward improvement in COVID-19 infection. On day 9, titers of SARS-CoV-Ab antibodies (N-IgG N and S-IgG; determined using the Abbot assay kit, Abbott, Abbott Park, IL, USA) remained unchanged from those of day 2, and there was no induction of immune response associated with the current infection. Remdesivir was administered for 10 days. Corona antigen levels on days 13 and 17 were 162 and 24 COI, respectively. Because no clinical evidence of COVID-19 worsening was noted, the patient was discharged on day 18. The patient had 1.0 COI 10 days after discharge and no relapse (Table 1).

## 3. Materials and Methods

### 3.1. COVID-19 Antigen Testing

We measured the COVID-19 antigen levels using the HISCLTM SARS-CoV-2 Ag Assay Kit (Sysmex, Kobe, Japan) with the HISCL-5000 analyzer (Sysmex, Kobe, Japan). The HISCLTM SARS-CoV-2 Ag assay kit details were obtained from a previously published reference [2]. In the SARS coronavirus antigen test, a labeled antibody that specifically reacts with the coronavirus antigen was used. In the chemiluminescence enzyme immunoassay (CLEIA), the luminescence intensity of the negative and positive samples, as well as the patient specimens, were assessed to determine the coronavirus antigen quantity. The amount of quantified coronavirus antigen is expressed as a COI value, which is set to 1 when the coronavirus antigen level reaches 3.65 pg/mL, and it is then converted into estimated copy numbers using a previously reported formula [2]. 

### 3.2. IgG Testing

The SARS-CoV-2 N-IgG and S-IgG levels were measured using the SARS-CoV-2 IgG assay kit (Abbott, Abbott Park, IL, USA) and SARS-CoV-2 IgG II Quant assay kit (Abbott, IL, USA), respectively, with the ARCHITECT i2000SR immunoassay analyzer (Abbott, IL, USA). 

## 4. Results

### 4.1. Ag and IgG Antibody Levels

Figure 1 and Table 2 present the trends in humoral and cellular immunity indices.

### 4.2. Cellular Immunity before and after the Onset of COVID-19 Infection

Table 2 shows the transition of the indicators that reflect the state of the immunity in the case.

## 5. Discussion

Liquid immunity to SARS-CoV-2 is primarily composed of IgM, which is transiently increased in the early stages of infection, and IgG, which is produced through class switch and maintained over time. The N-IgG is induced after natural infection [3], while the S-IgG levels are dramatically elevated after vaccinations [4]. In the present case, despite the S-IgG levels being elevated through vaccination and infection, the N-IgG levels notably remained low, even though the N-IgG levels typically increase after COVID-19 infection. There is limited knowledge regarding changes in antibody titers and the impact of anti-SARS-CoV-2 monoclonal antibodies on antibody titers in patients with hematopoietic disease receiving rituximab. The potential reasons for the lack of elevated N-IgG antibody levels, despite an effective immune response from the vaccine and Evusheld, including the anti-S (spike protein) antibody, were as follows: (1) the N-IgG levels did not increase because the N-protein was exposed for the first time in this infection, and (2) the Evusheld is believed to possess a relatively high capacity to induce antibodies; therefore, the S-IgG may still be positive despite the immunosuppressive effect of rituximab.

The persistence and recurrence of COVID-19 infection in patients who have undergone B-cell depletion therapy have posed challenges; however, no established method of vaccination or antibody therapy exists for these patients. The question of whether anti-SARS-CoV-2 monoclonal antibodies act directly or indirectly through the autologous immune system during infection remains controversial. In this case, the antigen levels in the antigen quantification test decreased rapidly after admission, indicating that the vaccine and monoclonal antibodies may have exerted an effect that was independent of IgG immunity. Furthermore, the absence of elevation in the N-IgG antibody titers could serve as a potential predictor of prolonged COVID-19 infection.

Table 2 presents the changes in the indicators of cellular immunity before and after COVID-19 onset. Because the serum IgG levels were maintained at approximately 600–700 mg/dL, immunoglobulin supplementation was not performed. The patient had a history of bendamustine use. The CD4-positive T cells decreased to approximately 200 counts/μL before the onset, decreased to 155 counts/μL 1 month after the infection, and subsequently recovered to the original level. A low CD4-positive cell count is a risk factor for prolonged viral shedding [5]. Therefore, in this case, prolonged COVID-19 infection was attributed to not only a deficiency in humoral immunity but also T-cell dysfunction.

In the PROVENT study [6], which investigated the efficacy and safety of Evusheld, the available information does not emphasize the extent to which the cases that did not demonstrate a positive response to SARS-CoV-2 nucleocapsid antibodies after receiving Evusheld included individuals with insufficient N antibody production caused by immunodeficiency or other underlying conditions. In addition, there is only one report on the use of Evusheld in patients with antineutrophil cytoplasmic antibody (ANCA)-associated vasculitis who received rituximab [7], with limited evaluations conducted on patients with lymphomas. The persistence and recurrence of the COVID-19 infection in patients undergoing B-cell depletion therapy have posed significant challenges; however, a well-established method of vaccination or antibody therapy for COVID-19 in these specific patients has not yet been developed. 

COVID-19 causes hyperglycemia through various mechanisms, including stress-induced hyperglycemia and direct or indirect effects on β-cells. It induces a universal biological stress response to infection and various mechanisms due to the COVID-19 infection [8]. It has been reported that the COVID-19 infection is associated with the development of new diabetes and the worsening of existing diabetes [8]. Compared with other viral infections, there is no high-quality evidence that elucidates how often COVID-19 causes severe metabolic disturbances such as HHS and diabetic ketoacidosis. Our understanding of the long-term impact on glucose tolerance is limited because of the short observation period after COVID-19 infection. Hyperglycemia itself is associated with unfavorable outcomes, and people with diabetes may experience prolonged symptoms, require readmission, or develop complications after recovery from COVID-19 [9]. Herein, the patient’s insulin demand temporarily increased and then decreased. Considering that at least 10% of infected patients experience long-term COVID [10], the question of whether COVID-19 may worsen vital and functional prognoses and accelerate aging in infected patients remains a subject for further investigation.

Prolonged hospitalization is not ideal for older patients because it limits their daily activities. However, if the patient is discharged from the hospital with a strongly positive antigen test, there is a risk of home infection [11]. Additionally, older patients with impaired hand dexterity, visual function, and cognitive function encounter difficulties in self-administering insulin, necessitating meticulous instructions for self-administration. To our knowledge, we encountered a case of the oldest COVID-19 and HHS patient who was cured after lymphoma remission. In this case, the patient demonstrated full independence in activities of daily living, with preserved motor function. However, her Mini-Mental State Examination score of 23 out of 30 points indicated mild cognitive impairment. The patient cohabitated with her husband and fell into Category II as per the “Glycemic Targets for Elderly Patients with Diabetes [12]”.

In the management of elderly individuals with diabetes, it is imperative to simplify treatment and establish a comprehensive support system for caregiving [13]. Given the observed decline in insulin secretion and the development of HHS, the initiation of insulin therapy was deemed necessary in this case. The patient received meticulous guidance on insulin injection techniques and self-monitoring of blood glucose levels. Although exercise therapy is typically employed to mitigate the decline in physical function during hospitalization, the present case involved a patient with COVID-19, which limited the extent of intervention. A brief assessment of the patient’s physical function was conducted during hospitalization, and exercise therapy was recommended for implementation post-discharge.

## 6. Conclusions

The impaired humoral response in lymphoma patients after anti-CD20 treatment implies the need for a justified different vaccination strategy for these patients. Additionally, the negative N-IgG titers in the immunosuppressed state may serve as an indicator of resistance to therapy. We hope that confirmation of the antibody levels throughout the treatment course of this case will provide valuable insights into preventive and therapeutic approaches for this patient population. The long-term clinical follow-up and biological monitoring of the immune response are necessary to determine the impact of lymphoma and its treatment on immunity and long-term prognosis in COVID-19-infected patients.

## Figures and Tables

**Figure 1 geriatrics-08-00110-f001:**
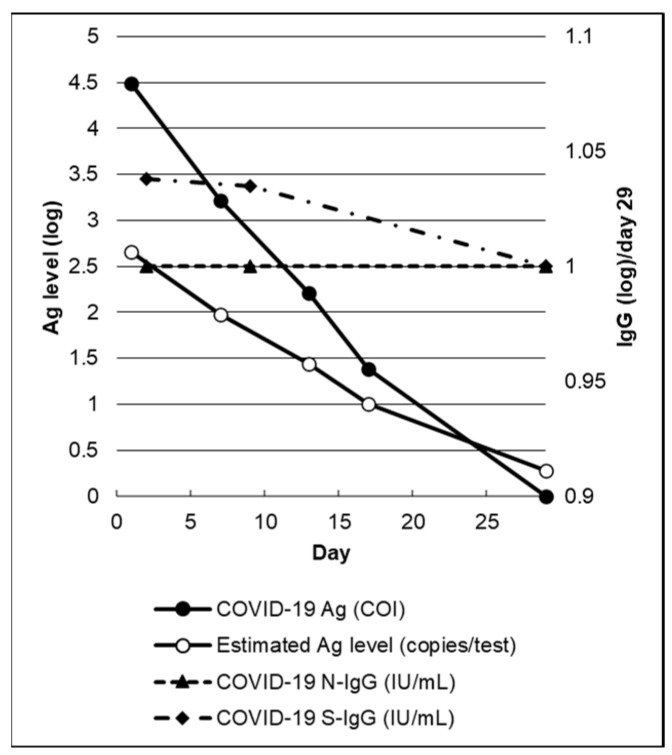
Serum Ag levels are shown on the left vertical axis. Serum IgG levels, referring to those on day 29, are shown on the right vertical axis. Both are presented in logarithmic form with a base of 10.

**Table 1 geriatrics-08-00110-t001:** Treatment course. The patient had been vaccinated and administered anti-SARS-CoV-2 monoclonal antibodies, but she tested negative for neutralizing antibodies upon admission. Though both neutralizing and spiking antibody levels were not persistently elevated, antigen levels progressively decreased during quantification testing. I.V., intravenous administration; S.C., subcutaneous injection; SSI, sliding scale insulin therapy. Changes in therapeutic dose and route of insulin administration were depicted in grayscale.

Day	1	2	4	7	9	10	11	13	17	29
WBC (counts/μL)	1400	2000	1800	2200	3100			2200		
CRP (mg/dL)	2.24	4.1	4.98	1.78	4.12			0.46		
Glu (mg/dL)	437	201		162	97			71		
COVID-19 Ag (COI)	30,802			1640				162	24.3	1
Ag level (copies)	450			95				28	10	2
COVID-19 N-IgG (IU/mL)		0.01			0.01					0.01
COVID-19 S-IgG (IU/mL)		25,172			24,364					17,339
Remdesivir (mg/day)	200	100								
Cefepime (g/day)			1							
Insulin	I.V.			S.C.	SSI				S.C.	

**Table 2 geriatrics-08-00110-t002:** Changes in the indicators of cellular immunity before and after the onset of COVID-19.

Time Since Onset, Month	−5	−2	0	1	5
Lymphocyte (counts/μL)	1375	1104	630	836	1183
CD4+ T cell (counts/μL)	221	191	-	155	222
IgG (mg/dL)	766	665	-	609	694

## Data Availability

Not applicable.

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
