# Peer review of "Prolonged COVID-19 Infection in a Patient with Complete Remission from Follicular Lymphoma with Hyperosmolar Hyperglycemic Syndrome"

_geriatrics, 2023, doi:10.3390/geriatrics8060110_

Round 1

Reviewer 1 Report

Comments and Suggestions for Authors

In this manuscript, Takuya Omura, et al. demonstrated that the impaired humoral response in lymphoma patient after anti-CD20 antibody treatment hampers the elevation in N-IgG antibody titers inducing prolonged COVID-19 infection. I think this is important and somewhat interesting, but there are some problems as indicated below.

Major point

1.     It is considered that prolonged COVID-19 infection is caused by not only the lack of humoral immunity but also T-cell dysfunction. In this case, the patient had a history of administration of bendamustine, and her cellular immunity may have been depressed at diagnosis of COVID-19. Did she have lymphopenia? The author should discuss more precisely these points and show additional data, such as the number of lymphocytes, CD4 positive T cells, serum IgG level, and so on.

2.     In this case, the patient also had hyperosmolar hyperglycemic syndrome (HHS) when COVID-19 developed. Is there the association between HHS and COVID-19? How does the author think? It is considered that diabetes cause immunodeficiency. The author should also discuss these points more precisely.

I believe that the authors must revise or at least discuss these issues more precisely to publish in Geriatrics.

I hope that my comment is useful for the improvement of the article.

Author Response

We sincerely appreciate your careful consideration and kind feedback. We have made the necessary revisions to address the issues raised. The detailed responses to your comments (reviewer 1) are attached as a Word file. Please see the attachment. Many thanks.

Best regards,

Takuya OMURA

Reviewer 2 Report

Comments and Suggestions for Authors

A paper by Omura T, et al., entitled “Prolonged COVID19 infection in a patient with complete remission from follicular lymphoma with hyperosmolar hyperglycemic syndrome” is a case report on an elderly patient who was vaccinated several times against SARS-CoV-2. Yet, the high levels of viral antigens.

No doubt, this case report should be seen by a scientific community. However, more details on what viral antigens were tested should be given. I mean, more details on methods should be given in the text.

After this, this case report might be accepted.

Comments on the Quality of English Language

No comments

Author Response

We sincerely appreciate your careful consideration and kind feedback. We have made the necessary revisions to address the issues raised. The detailed response to your comment (reviewer 2) is attached as a Word file. Please see the attachment. Many thanks.

Best regards,

Takuya OMURA

Round 2

Reviewer 1 Report

Comments and Suggestions for Authors

The authors answered all the questions raised by the reviewers and made the modifications accordingly in the manuscript.